# Severe Hypercapnia during Anaesthesia under Mechanical Ventilation in Two Paediatric Patients

**DOI:** 10.3390/ani13040663

**Published:** 2023-02-14

**Authors:** Anastasia Papastefanou, Eva Rioja

**Affiliations:** Optivet Referrals, Ltd., Havant PO9 2NJ, UK

**Keywords:** hypercapnia, hypoventilation, paediatric patient, mechanical ventilation, dynamic compliance, dead space

## Abstract

**Simple Summary:**

Mechanical ventilation is mandatory during ophthalmological surgeries when neuromuscular blocking agents are used. Ventilation of paediatric patients is more complicated because of their small size, the underdevelopment of their respiratory system and the lack of evidence-based guidelines for this age group. We presented two cases of severe hypoventilation during anaesthesia in paediatric dogs due to inadvertent ventilation mainly of the upper airways, and possible development of lung collapse. Both patients were ventilated using the same ventilator in a volume-controlled mode, and the actual level of hypercapnia was detected only when manual ventilation was used, and alveoli were properly ventilated. Adjustments in the volume-controlled ventilation failed to ventilate effectively, and manual ventilation was used for the remainder of the procedure until recovery. Both patients recovered well. Pressure-controlled ventilation, ventilators that take into consideration the compliance of the breathing system and the use of manual breathing, among other techniques, could contribute to better ventilation of those patients.

**Abstract:**

A 2-month-old male 1.56 kg Yorkshire terrier (Case No. 1) and a 3-month-old male 2.3 kg Jack Russell Terrier (Case No. 2) were scheduled for ophthalmological surgery under general anaesthesia and neuromuscular blockade. For both patients, volume-controlled ventilation (VCV) was used with set tidal volumes (V_T_) of 13 mL/kg and 20 mL/kg for cases No. 1 and 2, respectively. The type of ventilator used did not take into account the intrinsic compliance of the breathing system; therefore, a significant part of the delivered V_T_ was wasted in the expansion of the breathing system, and did not reach the patients, causing alveolar hypoventilation. Both cases developed low dynamic compliance (C_D_), and after a recruitment manoeuvre, EtCO_2_ of up to 116 mmHg and 197 mmHg were revealed for cases No. 1 and 2, respectively. The two cases had to be ventilated manually, using positive inspiratory pressures (PIP) of 20–25 mmHg, in order to improve alveolar ventilation and reduce the EtCO_2_, as adjustments to the VCV were ineffective. Both patients maintained an oxygen haemoglobin saturation between 94% and 100% throughout the procedure and they recovered well. Using a higher V_T_ from the beginning, to compensate for the compliance of the breathing system, or the use of pressure-controlled ventilation (PCV), could have potentially helped to avoid these two incidences of severe hypercapnia.

## 1. Introduction

Mechanical ventilation is necessary during anaesthesia to prevent and manage hypercapnia when the patient is hypoventilating or apnoeic for various reasons, such as the use of neuromuscular blocking agents (NMBA). The main goal of mechanical ventilation is to provide the appropriate minute ventilation ($\dot{V}$_E_) in order to meet oxygen requirements and eliminate carbon dioxide effectively when patients fail to ventilate themselves. Certain types of ophthalmological surgeries require the central position of the eye during general anaesthesia, and this can be achieved with the administration of NMBA. This makes the use of mechanical ventilation mandatory in these patients.

There are conventional and protective ventilation techniques, with the latter including low tidal volume (V_T_), low driving pressures and the application of positive end-expiratory pressure, with or without a lung recruitment manoeuvre [1]. Thus far, according to our knowledge, there are no studies in paediatric veterinary patients regarding optimal ventilation technique. Specific characteristics of respiratory physiology, such as immature respiratory control, increased collapsibility of the lower airways and reduced functional residual capacity, put patients at higher risk of hypoventilation or even respiratory failure under anaesthesia [2].

## 2. Case Report

Both presented cases were treated on the same day in the following order.

Case No. 1

A 2-month-old, male 1.56 kg Yorkshire terrier was admitted for cornea-conjunctival-transposition (CCT) surgery of the right eye, after suffering a full thickness corneal trauma and iris prolapse following a cat scratch. The dog was healthy on presentation, and its biochemistry was unremarkable. The haematocrit was 28.3% on a blood sample taken after induction of anaesthesia. The dog was premedicated with a combination of methadone (0.2 mg/kg) (Methadyne, Jurox, Crawley, UK), midazolam (0.2 mg/kg) (Hameln pharma LTD, Gloucester, UK) and alfaxalone (2 mg/kg) (Alfaxan/Multidose, Jurox, Dublin, Ireland) intramuscularly (IM). Two cannulas were placed, one into the right cephalic vein and another one into the left saphenous vein. For induction of anaesthesia, a total of 2 mg/kg of alfaxalone was administered to effect through the left cephalic vein until endotracheal intubation could be achieved without a coughing response. The patient’s trachea was intubated using a cuffed polyvinyl chloride (PVC) endotracheal tube (ETT), with an internal diameter (ID) of 4 mm. The cuff of the ET tube was inflated using the minimum occlusive volume (MOV) technique, consisting of inflation of the cuff with air, using a regular syringe until there was no audible air leakage noise when applying positive airway pressure of 20 cmH_2_O with a manual breath. Anaesthesia was maintained with sevoflurane (Sevohale, Chanelle, Loughrea, Ireland) in 100% of oxygen administered through a Mapleson D breathing system, with a fresh gas flow rate of 1.5 L/min. During anaesthesia, heart rate and rhythm, haemoglobin oxygen saturation (SpO_2_), oscillometric blood pressure, oesophageal temperature and capnography were continuously monitored and recorded every 5 min using a multiparameter monitor (Datex-Ohmeda, GE, Helsiniki, Finland). Spirometry was also continuously monitored, using the same monitor, but was recorded every 15 min. For spirometry, a Pedi-Lite flow sensor (Pitot tube, Datex-Ohmeda/GE Healthcare, Helsiniki, Finland) was attached between the heat and moisture exchange (HME) filter and the breathing circuit. The sampling tube of the capnograph (side stream) was attached to the port of the HME filter, with a sampling rate of 200 mL/min. Sixty-five minutes after induction, atracurium (0.2 mg/kg) (Tracrium, Aspen Pharma, Dublin, Ireland) was administered through the left saphenous vein. Until then, the animal was breathing spontaneously. An additional dose of atracurium (0.2 mg/kg IV) was administered 50 min after the initial dose. Neuromuscular activity was monitored at the level of the right peroneal nerve with acceleromyography, using a Train-of-Four (TOF) stimulation pattern (Stimpod NMS20450X, Xavant Technology, Pretoria, South Africa). The animal was ventilated with volume control ventilation (VCV) using a calculated V_T_ of 13 mL/kg (20 mL total), respiratory rate (RR) of 10–15 bpm and zero positive end-expiratory pressure (ZEEP), using a non-rebreathing system set-up (Merlin, Vetronic Services LTD, Devon, UK). When ventilation was initiated, the “low inspiratory pressure” alarm sounded, and the inspired and expired V_T_ were 0 mL on the spirometer. Therefore, the V_T_ setting on the ventilator was increased gradually from 20 to 50 mL until the spirometric measurements recorded on the monitor were the following: expired V_T_ of 14 mL, positive inspiratory pressure (PIP) of 13 cmH_2_O, positive end-expiratory pressure of 0 cmH_2_O (ZEEP) and the dynamic compliance (CD) of the patient was 1.2 mL/cmH_2_O. The EtCO_2_ at that moment was between 37–40 mmHg, and there were no abnormalities on the capnograph. In order to improve the C_D_, a stepwise alveolar recruitment manoeuvre was attempted, which consisted of the addition of 5 cmH_2_O of PEEP incrementally every 5 breaths until the PEEP reached 10 cmH_2_O [3]. Afterwards, the PEEP was reduced to 5 cmH_2_O and maintained at that level. As this manoeuvre did not improve the C_D,_ the configuration of the system was changed to a rebreathing system, and a different recruitment manoeuvre was performed, consisting of a manual breath to achieve a PIP of 20 cmH_2_O, maintained for 20 s. This resulted in an improvement of the C_D_ to 1.9 mL/cmH_2_O, and an EtCO_2_ measurement of 116 mmHg. At this moment, the VCV changed to manual ventilation (MV) at a RR of 30–35 bpm, aiming for a PIP of 15–20 cm H_2_O, to try to improve alveolar ventilation. Approximately 5 min after MV was initiated, the EtCO_2_ dropped to 58 mmHg, but it would rapidly increase again to levels of 99 mmHg when MV was stopped and VCV initiated; therefore, MV was maintained until the end of anaesthesia. There were no recordings of the C_D_ after that point on the anaesthesia record. The SpO_2_ ranged between 94–100 mmHg, and all the other parameters were within acceptable limits during the whole anaesthetic episode. The TOF readings during surgery ranged between 0/4 and 1/4, until it was reversed with the use of neostigmine (0.05 mg/kg) combined with glycopyronium (0.01 mg/kg) (Glycopyronium Bromide and Neostigmine Metilsulfate, Mercury Pharma, London, UK) IV. When the patient started breathing spontaneously again, the EtCO_2_ dropped to levels below 60 mmHg, and extubation was carried out 5 min after the end of inhalational anaesthesia. The recovery was smooth and uneventful. The anaesthesia duration was 145 min, and the total time the patient was on MV was 45 min.

Case No. 2

A 3-month-old male 2.3 kg Jack Russell Terrier was admitted for corneal suturing following a cat scratch. The dog was otherwise healthy on presentation, and there was no other relevant clinical history. He was premedicated with methadone (0.2 mg/kg), midazolam (0.2 mg/kg) and alfaxalone (2.5 mg/kg) IM. Two cannulas were placed, one into the right cephalic vein and another in the right saphenous vein. For induction of anaesthesia, a total of 2 mg/kg of alfaxalone was administered to effect through the right cephalic vein until endotracheal intubation could be achieved without a coughing response. The dog’s trachea was intubated using a cuffed PVC ETT with a 4 mm ID. Inflation of the cuff was performed using the same technique as in the previous case. Anaesthesia was maintained with sevoflurane in 100% of oxygen administered through a Mapleson D system. During anaesthesia, heart rate and rhythm, SpO_2_, oscillometric blood pressure, oesophageal temperature, capnography and spirometry were monitored and recorded similarly to case 1 using the same multiparameter monitor. Forty minutes after the induction of anaesthesia, atracurium (0.2 mg/kg) was administered through the right saphenous vein. Until then, the dog was breathing spontaneously. Neuromuscular activity at the level of the peroneal nerve was monitored in the same manner as in the previous case. Just before atracurium administration, the Mapleson D breathing system was changed for a rebreathing system, and ventilation was maintained using the same ventilator with VCV under the following initial settings: calculated V_T_ of 20 mL/kg (45 mL total), respiratory rate of 10–15 bpm and ZEEP. The spirometer of the monitor was reading an expired Vt of 20 mL and a PIP of 8 cm H_2_O. During anaesthesia, the BP was within normal limits, and the SpO_2_ was between 98–100%. The EtCO_2_ gradually began to increase above 60 mmHg, and adjustments to the RR and small increases to the V_T_ were not effective in correcting this. Twenty-five minutes after the atracurium administration, the EtCO_2_ suddenly reached 96 mmHg, due to spontaneous respiratory efforts. At that moment, the PIP was 9 cmH_2_O, PEEP was 0 cmH_2_O, expired V_T_ was 25 mL and the C_D_ was 2 mL/cmH_2_O, as measured with the spirometer of the monitor. An alveolar recruitment manoeuvre with manual breathing to achieve a PIP of 20 cmH_2_O was performed and maintained for 20 s, following which a PEEP of 5 cmH_2_O was applied. After that, the monitor was measuring an EtCO_2_ of 197 mmHg. As the neuromuscular block (NMB) started to wear off—the TOF ratio was 0.66—the patient began to fight the ventilator, the ventilation became erratic, and the driving pressure (DP) (DP = PIP − PEEP) was between 12 and 25 cmH_2_O. Due to the difficulty in mechanically ventilating the patient, the VCV and PEEP were stopped, and MV was performed until the end of anaesthesia. The NMB was reversed with neostigmine (0.05 mg/kg) combined with glycopyronium (0.01 mg/kg) IV (Glycopyronium Bromide and Neostigmine Metilsulfate, Mercury Pharma, UK), 5 min after MV was initiated. Unfortunately, there were no recordings of the spirometer measurements during MV, but the EtCO_2_ dropped to 65 mmHg within 5 min of MV and was 45 mmHg just before the recovery. Extubation occurred 15 min after the end of inhalant anaesthesia, and the patient recovered well and uneventfully. The duration of anaesthesia was 105 min, and the patient was on MV for a total of 20 min.

## 3. Discussion

When using controlled mechanical ventilation during anaesthesia in small animals, the most used modes are VCV and pressure control ventilation (PCV) [4]. Acceptable V_T_ settings reported for adult dogs are 8–15 mL/kg, with a PIP of 8–15 cm H_2_O when no lung pathology is present [5,6]. In humans, the optimal V_T_ is 6–8 mL/kg for adults, and 4–8 mL/kg of ideal body weight (IBW) for paediatric patients [7]. A low V_T_ is used in humans because it has been proven that lung protective strategies reduce the risk of postoperative pulmonary complications [1]. In veterinary medicine, a recent study in healthy dogs, positioned in lateral recumbency and without an NMBA being used, evaluated the clinical feasibility of a V_T_ of 8 mL/kg; the study concluded that this volume was sufficient to maintain normocapnia; however, this volume was not compared with other V_T_ values [8]. The use of a low V_T_ in dogs is still controversial, and there is evidence that volumes of 15 mL/kg are considered more appropriate to avoid hypercapnia [9,10,11]. This is in part due to dogs having a greater airway dead space compared to humans; for that reason, a higher V_T_ setting is normally needed in order to achieve adequate alveolar ventilation [9,12,13]. In one case report, the V_T_ achieved in a pressure-controlled ventilated paediatric canine patient was between 11–18 mL/kg during 116 h of mechanical ventilation [14]. Ventilation of paediatric patients, especially when they are less than 2 kg of body weight, can prove to be challenging, and in human medicine these patients have traditionally been ventilated with PCV. The main advantages of PCV compared to VCV include the following: (i) it compensates for small leaks, allowing delivery of the desired tidal volume; (ii) the pressure required to overcome resistance is rapidly generated and then maintained for the entire inspiratory time, favoring lung recruitment; (iii) the flow of gas is high, initially, and then decreases gradually during inspiration (descending or decelerating flow), hence improving the distribution of gas within the lungs, reducing alveolar dead space and increasing oxygenation, especially in non-homogeneous lungs [7,15]. The main disadvantage of PCV is that a stable V_T_ is not always achieved, due to potential changes in lung-thorax compliance or obstructions at the level of the ETT, which will cause a reduction in alveolar ventilation.

Protective ventilation techniques involve the use of low V_T_, alveolar recruitment manoeuvres (RMs) to re-open collapsed alveoli (“open lung concept”), and the use of PEEP afterwards to maintain the alveoli open in order to avoid re-collapse [7]. In humans, there are different established RM techniques: either a sustained inflation of the lungs up to a PIP of 40 cm H_2_O, maintained for 10–40 s, or a gradual increase in the PIP up to 40 cm H_2_O by progressively adding PEEP (“stepwise manoeuvre”) [16]. In veterinary medicine, both techniques have been studied in healthy anaesthetised dogs and proven to be effective to re-open atelectatic alveoli [17]. Although it may cause greater depression of the cardiovascular system, the sustained inflation technique is easier to perform [10]. The stepwise RM improves dynamic compliance and oxygenation in healthy dogs [3], but it requires more sophisticated equipment and better-trained personnel. To our knowledge, no studies have been performed in veterinary paediatric patients; however, in human paediatric patients, similar techniques as for the adult population have been used [18].

As mentioned above, part of the indications for an RM is to improve lung compliance by opening atelectatic alveoli. In veterinary patients, there is little evidence regarding the optimal compliance during MV. In one study in adult medium-sized dogs, it was proposed to use the following equation as a tool to monitor the dynamic compliance of the respiratory system (Crs) during anaesthesia: Crs = BW + 9 [19]. In healthy paediatric human patients, static respiratory system compliance was reported to be between 14.7 and 20.6 mL/cm H_2_O/kg [20]. According to Lump (2017) [21], lung compliance increases a little with age, and larger animals have greater compliance compared with smaller ones.

When using VCV, the V_T_ displayed on the control panel of the ventilator used in these two cases represented the volume of gas the ventilator was set to deliver. However, not all this set volume reached the patient, as a proportion of it was lost due to the distensibility or intrinsic compliance of the breathing circuit tubing (C_T_). During the inspiratory phase, the gas becomes compressed within the breathing system and the tubing gets distended, which causes a reduction in the V_T_ that reaches the patient’s lungs. The C_T_ can significantly affect the V_T_ that the patient receives, especially in very small patients, such as neonates and paediatric patients, as the set V_T_ is already very small; therefore, a greater proportion of this set V_T_ is “absorbed” by the tubing [7]. Most modern ventilators calculate the C_T,_ which can vary between 2–5 mL/cm H_2_O, meaning that if the PIP is 10 cm H_2_O, a total of 20–50 mL of the set V_T_ will not reach the patient. For a patient weighing <2 kg, this is a very significant loss. Additionally, in human medicine there is evidence that airway dead space per kg of body weight decreases with age [2]. If this is also true in veterinary paediatric patients, it means a higher V_T_ would need to be used in order to reach the alveoli and avoid ventilating only the conducting airways.

In the presented cases, anaesthesia monitoring was performed by an anaesthesia-dedicated nurse trained by an anaesthesia specialist (ERG) whose telephonic assistance was requested when complications occurred. Furthermore, in both cases the access to the head and thorax of the dogs to evaluate possible cuff leaks and thoracic excursions was limited, due to the required positioning for surgical access. As a result of their small size, the dogs were completely covered by the surgical drapes, and the head was placed under the microscope, which added difficulty in evaluating these patients. Even small movements under the surgical drapes had a great impact during ocular surgeries.

In the first case, the order of the events and actions taken were as follows: initially, the low-pressure alarm of the ventilator sounded and prompted the nurse to increase the set V_T_ on the ventilator and request assistance. When enough V_T_ was reaching the patient (V_T EXP_ = 14 mL, measured by the spirometer), the C_D_ was calculated by the spirometer to be 1.2 mL/cmH_2_O (Appendix A). A stepwise manoeuvre was initiated, up to 10 cmH_2_O, only to avoid barotrauma in the paediatric lung, which did not increase the compliance. The circuit was changed to a rebreathing system, in order to provide a sustained inflation to a PIP of 20 cmH_2_O for 20 s, which increased the C_D_ to 1.9 mL/cmH_2_O. At that point, the ETCO_2_ increased from 37 to 116 mmHg. Adequate ventilation allowing a reduction in ETCO_2_ was only achieved when MV was started, possibly because it is more similar to PCV and allowed better alveolar ventilation.

For the second case, after having the experience of case No. 1, a rebreathing configuration and a preset V_T_ of 20 mL/kg were used from the beginning of mechanical ventilation, in order to be able to perform a sustained inflation RM if needed and taking into consideration the loss of V_T_ due to the C_T_, respectively. Again, as in the first case, a significant volume was lost due to the C_T_ (Appendix A), and the patient was under-ventilated with an expired V_T_ of only 20 mL. The developed hypercapnia and low C_D_ triggered the monitoring team to perform an RM, which unmasked a severe hypercapnia that was also managed successfully with MV.

It is obvious that the V_T_ of 13 mL/kg that was set on the ventilator at the beginning of anaesthesia in case No. 1 was not sufficient to offset the C_T_ and reach the alveoli, and this was because this type of ventilator does not take into consideration the compliance of the breathing system. This type of ventilator has a screen panel that measures the V_T_ given and the PIP achieved after each delivered breath, but does not provide the expired V_T_. These measurements of inspired V_T_ and PIP are not obtained at the patient end, but at the ventilator end; therefore, there is a significant discrepancy between the parameters measured by the ventilator and those measured by the spirometer attached to the ETT of the patient, as was seen in both cases (Appendix A). Although the PIP measured by the ventilator was not reported here, from our experience, it also routinely reads higher compared to spirometry readings.

In the presented cases, the measured C_D_ at the beginning of mechanical ventilation was very low. Some of the factors that reduce lung compliance during anaesthesia include small lung volume, posture, absorption atelectasis and reduced tone of bronchial smooth muscles [21]. In these cases, both dogs were receiving a low V_T_ that was not reaching the alveoli, although not for a very long time. They were both in dorsal recumbency, which contributed to a reduced functional residual capacity (FRC) [21] and were paralysed with an NMBA, which likely reduced the FRC further by reducing respiratory muscle tone. Additionally, absorption atelectasis was highly likely in both cases, as the dogs were receiving 100% oxygen. A sustained inflation RM cannot be performed when this ventilator is set up to a non-rebreathing configuration. For this reason, in case No. 1, a stepwise RM was performed initially, reaching a maximum PEEP of 10 cmH_2_O instead of 20 cmH_2_O, which is the recommended maximum PEEP in adult patients, in order to prevent barotrauma in this paediatric patient. However, it proved to be inefficient, and a decision was made to change the set-up of the ventilator to a rebreathing configuration, and to perform a sustained inflation RM. This revealed a very high EtCO_2_ of 116 mmHg, which was probably the result of delivering an inadequate V_T_ and possibly of the RM itself, as during these manoeuvres the EtCO_2_ increases transiently. There are different potential causes of impaired blood–gas exchange, including the following: alveolar hypoventilation, alveolar dead space (high V/Q ratios), venous admixture (low V/Q ratios), diffusion impairment and right-to-left shunt (V/Q = 0). The main cause of hypercapnia in the presented cases was alveolar hypoventilation, as discussed earlier. Areas with low V/Q ratios (e.g., partially collapsed alveoli) could also have contributed to the hypercapnia and would have been accompanied by high alveolar-arterial O_2_ differences; unfortunately, we did not measure arterial blood gases in these cases. Atelectasis and right-to-left shunt would have resulted in reduced PaO_2_ and even greater alveolar-arterial O_2_ difference, which could have caused haemoglobin desaturation. However, these patients were maintained on a high FiO_2_, and therefore desaturation is not likely to happen until at least 35–40% of shunt occurs [22]. The RM would help to re-open the collapsed alveoli, increasing the area of efficient blood–gas exchange, allowing for the accumulated CO_2_ to be eliminated. Additionally, increasing the V_T_ without an initial RM could cause over-distension of alveoli that are already opened, potentially increasing alveolar dead space.

Many ophthalmological cases require the use of NMBA, in order to maintain the eye in a central position. Mechanical ventilation is mandatory in those cases. Most of the time, mechanical ventilation starts after the administration of the NMBA, as in the presented cases. Starting mechanical ventilation before the administration of the NMBA could potentially allow time to adjust the settings; however, spontaneous efforts of the patient may make it difficult to adjust the settings, causing patient-ventilator asynchrony. Monitoring the thoracic wall excursions before the start of the surgery could have helped to assess the effectiveness of ventilation in these cases, in conjunction with the recorded V_T_. This is especially important when spirometry is not available.

One event that went unnoticed was the discrepancy between inspired and expired V_T_, as seen in Appendix A and, to a lesser extent, in Appendix A. There was a difference of 8 mL in the first case that could have been the result of a leak at the level of the ETT or the breathing system, but this is unlikely as the system was checked for leaks before the beginning of anaesthesia. Moreover, the aspiration flow rate of the side stream capnograph on this type of monitor is 200 mL/min, which could also have contributed in part to this discrepancy.

An alternative approach instead of performing MV when hypoventilation was detected would have been to adjust the ventilator settings to a higher V_T_ and allow a higher PIP to be reached, and apply PEEP after the RM; however, the decision to change to MV was made because the monitoring team was more confident with this technique, and this would result in faster reduction in the EtCO_2_, as it was extremely high. The use of PCV instead of VCV would have been another option to deliver more efficient breathing, due to the longer period of high pressure, and to overcome any potential leak; however, the monitoring team was not familiar with this type of ventilation. In both cases, the decision of the monitoring team was to use a conservative approach initially, due to the age of the patients, and to set a low V_T_ of no more than 10 mL/kg on the ventilator. Unfortunately, the low V_T_ delivered resulted mainly in the ventilation of the airway dead space, which led to severe hypercapnia. Fortunately, adequate ventilation was achieved quickly with MV, and the EtCO_2_ decreased to acceptable levels before the end of anaesthesia. As the FiO_2_ was close to 1 (100% inspired oxygen), the haemoglobin saturation was normal in both cases despite severe hypercapnia. Both animals recovered quickly and well, without any untoward neurological signs.

## 4. Conclusions

In paediatric small animals, it may be more efficient to ventilate with pressure control ventilation rather than volume control ventilation, especially if a ventilator that does not compensate for C_T_ is used. If VCV is used, the set tidal volume needs to be adjusted to take into consideration lost volume due to the compliance of the breathing system. Additionally, a manual breath up to a PIP of 20 cmH_2_O could be delivered a few minutes after the beginning of mechanical ventilation to detect possible hidden hypercapnia due to dead space ventilation. This may allow adjustment of the settings accordingly by either increasing the set tidal volume or decreasing apparatus dead space, if possible. Finally, the addition of PEEP from the beginning of ventilation and F_i_O_2_ < 1 from the start of anaesthesia could have reduced the atelectasis formed and maintained better compliance. When assessing the adequacy of ventilation during mechanical ventilation in a small patient, one should not forget that apparatus dead space adds to the airway dead space. The closer the sampling line of the capnograph to the patient’s mouth, the higher the degree of accuracy. Moreover, the spirometer’s measurements, when available, as opposed to the ventilator’s readings, are more accurate, and should be used to guide ventilation. Monitoring the thoracic excursions can provide valuable information about appropriate lung inflation, especially at the beginning of mechanical ventilation. Unfortunately, ophthalmological cases have a unique setting, making visual monitoring and access to the patient difficult. Arterial blood gas analysis, albeit challenging in a small paediatric patient, could provide valuable information, not only when discrepancies occur and hypoventilation ensues, but also from the beginning of ventilation, as it will help to assess blood–gas exchange. The use of this specific type of ventilator, while it is designed to allow a large range of tidal volumes and provides information that helps to assess the efficacy of ventilation, did not prevent the occurrence of the complications described above.

With this case report, we would like to highlight the complexity of ventilating paediatric patients undergoing ophthalmological procedures and paralysed with an NMBA, using a ventilator that does not have the technology to compensate for C_T_. Vigilance and adaptation of the standards of care and management are necessary. Initiation of mechanical ventilation before placement of the surgical drapes, in order to monitor chest excursions; calculation of the initial V_T_ incorporating the lost volume within the breathing system; and the use of appropriate monitoring techniques, such as spirometry, capnography and blood gas analysis, could help to avoid similar complications.

## Data Availability

The presented case report is not a study, and no data were collected.

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
