# Peer review of "Severe Hypercapnia during Anaesthesia under Mechanical Ventilation in Two Paediatric Patients"

_animals, 2023, doi:10.3390/ani13040663_

Round 1

Reviewer 1 Report

Review
animals-2130034

Severe hypercapnia during anaesthesia under mechanical ventilation in two paediatric patients

This case report discusses 2 anesthetic episodes where significant hypercapnia occurred after the administration of neuromuscular blocking agents to small dogs undergoing ophthalmic surgery.  Initially, the dogs were being ventilated using a microprocessor-controlled ventilator than can be used in pressure cycled, volume cycled or time cycled modes and allows a maximum inspiratory to be set.  The dogs were initially anesthetized using a non-rebreathing system using volume-cycled settings and then were switched to rebreathing system as part of the response.

The first dog was anesthetized and endotracheally intubated and attached to the non-rebreathing system with ventilator in volume cycled mode. The size of the endotracheal tube or kind (cuffed or non-cuffed) is not indicated and there is not a statement that the airway was sealed.  In addition, the fresh gas flow rate (critical to a nonrebreathing system) is not indicated. An appropriate tidal volume (20 ml) was set initially but the ventilator computer system indicated a low inspiratory pressure, so the tidal volume was increased to 50 ml (33 ml/kg) and this silenced the alarm.  This tidal volume appears excessive with the potential to cause barotrauma.   The authors do not indicate whether the chest wall was moving appropriately at the level of ventilation set.

At the same time, the expired tidal volume indicated by the ventilator was 14 ml.  Expired tidal volume should approximately equal the delivered tidal volume which calls to question the integrity of the ventilator/anesthetic circuit combination and points to the possibility of a leak in the system.

The computer ventilator calculates a value reported as dynamic compliance.  The computer screen indicated a value purported to represent low dynamic compliance.  In response, the anesthetists instituted a recruitment maneuver via positive end expiratory pressure.  When the compliance value did not change and the end-tidal CO2 increased to a dangerous level (116-142 mmHg).

The second case has a similar presentation, and the treatment and outcome were similar.

In the reviewer’s experience, a level of end-tidal CO2 of this level is a machine/operator problem rather than a patient problem.  The fact that the cases occurred on the same day adds some credence to this possibility. The concerns center on rebreathing of expired gas due to a circuit malfunction (sticking valves), an inadequate fresh gas flow rate, a large leak in the system or potentially, a malfunction of the ventilator.  Expired CO2 absorbent is a circle system is an additional source but would be unlikely to change end-tidal CO2 as rapidly as in reported here.  Recruitment maneuvers have been reported to cause increases in PaCO2 so the rationale behind their being used is questioned.

In both cases, the operators shifted from the volume cycle ventilator to manual controlled ventilation, which the reviewer assumes is equivalent to pressure cycled ventilation (the patient was bagged to a given inspiratory pressure).  If leaks were present, this would overcome a leak whereas a volume cycled ventilator would not.

In the reviewer’s experience inherent issues with ventilatory compliance are rare in dogs, especially small dogs.  Causes external to the thorax (distend abdomen for example) would be a more likely cause.  In such an instant, recruitment maneuvers would be detrimental to venous return and cardiac output.

For the authors to sort through these cases, it would be useful to construct a table with the variables reported by the ventilator to determine what the sequence of events.

Author Response

We would like to thank you for revising our manuscript and giving us the correct direction to improve it.

We will try to answer all your comments as clearly as we can.

1st paragraph: You comment that the dogs were switched to a rebreathing system as part of the response. Both dogs were breathing through a Bain system when they were breathing spontaneously. Dog No. 1: initially he was mechanically ventilated through the specific configuration of the Merlin ventilator with a non-rebreathing set-up but switched to a rebreathing set-up during the procedure when ventilation was ineffective. Dog No 2 was breathing spontaneously through a Bain system and was mechanically ventilated through the rebreathing setup from the beginning of mechanical ventilation. We believe that this information is clearer now, after the changes.

2nd paragraph: Thank you for highlighting the missing information regarding the ETT tube and the fresh gas flow rate. We made the appropriate changes.

The initial tidal volume of 20 ml and later changed to 50 ml was the volume indicated on the screen of the ventilator. The ventilator doesn’t measure at the level of the ETT, like the spirometer. The “real” tidal volume that the patient was receiving initially was not recorded (likely 0), and afterwards, it was 22 ml as measured by the spirometer. This is the volume that was reaching the patient, and we believe this was the main problem in these presented cases; the lost volume into the breathing system due to the intrinsic compliance of the tubing. For this reason, we believe there was no risk of barotrauma. Also, the PIP when the tidal volume was increased to 50 ml on the ventilator, was 13 cmH2O, which should not cause barotrauma.

Thank you for your comment about appropriate chest movements, we didn’t check this as we did not have access to the thorax of the patients, but we agree this is something that could be checked especially when mechanical ventilation is started. This has now been added to the manuscript.

3rd paragraph: The expired tidal volume of 14 ml was measured by the spirometer. Indeed, there was a discrepancy between inspired and expired volumes measured by the spirometer (22/14 mls and 25/20 mls, in cases No 1 and 2, respectively), indicating that possibly there was a leak, especially in case No 1, where the discrepancy was 8 mls. We believe some of the lost volume could have been due to the high aspiration flow rate of the side stream capnograph (200 ml/min), which could account for approximately 3 mls, but there are still 5 mls lost that could have been certainly been due to a leak. We have now made a comment about this in the text.

4th paragraph: The dynamic compliance that we were relying on for our clinical decisions was the one measured by the spirometer. Unfortunately, we don’t have the dynamic compliance values measured by the ventilator. We usually don’t take them into consideration as the calculation is based on the set tidal volume and pressures measured by the ventilator, which can be very different to the ones measured by the spirometer, as discussed in the manuscript.

5th paragraph: The capnography didn’t indicate any signs of rebreathing of CO2. We agree that there was a leak in the system but not as large as 36 ml. As we mentioned above, 50 ml was the ventilator measurement and 14 ml was the spirometer measurement. The actual leak, as indicated by the difference between the inspired and expired tidal volumes was 8 mls (22-14 mls), and as explained above possibly only 5 mls constituted the real leak. We agree that a recruitment manoeuvre temporarily increases the ETCO2 but after opening the closed alveoli the elimination of the accumulated CO2 happens very quickly. We have added a comment about this in the discussion.

6th paragraph: We have added this to the discussion.

7th paragraph: We agree that changes in compliance during anaesthesia are much more profound in large animals than in small animals. However, dorsal position, use of NMBA, different breeds, like brachycephalics and the use of 100% of oxygen all can contribute to a reduction of compliance (De monte et all, 2013; Lutchman and Rutherford, 2019).

8th paragraph: We have now added two tables that we hope to add to the description of the cases.

De Monte, V., Grasso, S., De Marzo, C., Crovace, A. and Staffieri, F., 2013. Effects of reduction of inspired oxygen fraction or application of positive end-expiratory pressure after an alveolar recruitment manoeuvre on respiratory mechanics, gas exchange, and lung aeration in dogs during anaesthesia and neuromuscular blockade. American Journal of Veterinary Research74(1), pp.25-33.

Lutchman, A. and Rutherford, L., 2019, April. Prevalence of pulmonary changes detected using computerised tomography in brachycephalic breeds compared to mesaticephalic breeds. In BSAVA Congress Proceedings 2019 (pp. 495-495). BSAVA Library.

Reviewer 2 Report

Dear Authors, thank you for submiting clinical cases that are always interesting and that bring great discussion potentials.

I have some comments and questions concerning both case descriptions and the following discussion.

Ventilation modes, manoeuvres and monitroing measurements are well described however I feel some more clinical informations are missing and need also to be part of the discussion. Intubation information is lacking (cuffed ETT ID in mm?)

Technical questions: the spirometry measurements were done with the datex monitor. I'm guessing a spirometry flow sensor was therefore hooked to the endotracheal tube. Where was the sampling done for the gas analyser? The spirometry sensor adds significant deadspace to the circuit and can interfere with CO2 measurement depending on where the sampling point is comparaed to the ETT connector. On the non rebreathing circuit, the fresh gas flow can also impact that measurement.

Measures of respiratory pressures and compliance are interesting but reading your case descriptions, what is lacking from my point of view is the procedure allowing to assess the adequacy of ventilation. Was the chest excursion checked, was there some visible abdominal mouvement with the respiratory cycles witnessed.

The Merlin ventilator gives also some tidal volume informations. Were those compared to the datex spirometry data?

The fresh gas flow is not described either. It is part of the information needed as it can influence the tidal volume delivered by the ventilator.

All those aspects/informations should be added to the case description when available and part of the discussion if not available.

Your discussion would gain to have a part on procedure to assess the adequacy of ventilation while the anmal is on IPPV, and also the pratical aspect of using IPPV during NMBA. Is starting IPPV before atracurium injection and having the time to asses its effects even before the surgical drapes are covering the dog a practice to implement so it is easier to fine tune the settings. The use and here the lack of use of blood gas analysis should also be part of the discussion.

Concerning the part of the discussion on recruitment manoeuvers, it is well written and referenced, however you do not really discussed why they were needed in those cases. What are the possible causes of the lung collapse in those 2 tiny patients? Were there any other reason than an inadequate IPPV setting? Was the positioning of the animal or of the surgeon's hands/arms/instruments in any possible way hindering respiratory mouvements?

I also think that a discussion on pros and cons on VCV versus PCV for tiny patients would be relevant.

The conclusion should emphasize on how to assess the adequacy of IPPV settings and special aspects when considering tiny patients.

Best regards

Author Response

Thank you very much for the constructive comments. We took them all into consideration and we hope that we have improved the description and the following discussion in the revised manuscript.

3rd paragraph: Thank you for pointing out the missing information regarding the ETT. We have now added this information.

4th paragraph: We have added all the information about the position of the pedi-light, the sampling line for the gases and the suction rate, as well as the FGF, in the revised text.

5th paragraph: Unfortunately, we didn’t have access to the thorax because the patients were very small and the chest of the animals was very close to the surgeon under the drapes. Any movement under the surgical drapes in these cases could have a great impact on the surgery that is being done under the microscope. Nevertheless, we have added this information in the manuscript and in the conclusions recommend that this is done. Additionally, we have added two tables describing the different tidal volumes and airway pressures, as well as the compliance at different time points. Also, please see the relevant changes in the text.

6th paragraph: We had mentioned originally in the description of the case the difference in the tidal volumes between the screen of the ventilator and the spirometer. For example, for case No 1 the tidal volume setting on the ventilator was 50 ml, but the measured tidal volume on the spirometer was only 14 ml. The two tables we have added will help to clarify any confusion.

7th paragraph: Thank you for pointing this out to us. It has now been added to the text.

9th paragraph: Regarding the assessment of ventilation while the animal is under mechanical ventilation, we have commented in the discussion that a manual breath shortly after the beginning of mechanical ventilation would help to understand if there is alveolar hypoventilation (the EtCO2 would increase) or if the delivered tidal volume was adequate. (Note: we have changed the terminology in the text from anatomical to airway dead space after the recommendations of the third reviewer). We have added more in the discussion of the revised manuscript regarding mechanical ventilation and assessment of ventilation, plus the use of mechanical ventilation, NMBA and arterial blood gases. We don’t tend to ventilate patients that breathe spontaneously and have acceptable ETCO2 because they will most likely fight the ventilator and this causes patient-ventilator asynchrony, while after the administration of the NMBA ventilation can be adjusted better.

10th paragraph: Please see the explanation in the revised text. Thank you for highlighting it.

11th paragraph: Please see the revised text.

12th paragraph: Please the revised text.

Author Response

Thank you very much for the constructive comments. We took them all into consideration and we hope that we have improved the description and the following discussion in the revised manuscript.

General:

We have requested for the manuscript to be proofread by a native speaker.

Simple summary

  1. The journal requires lay language to be used in the simple summary. However, we agree that hypoventilation is more accurate. We have made the appropriate changes.

  1. We have made the necessary changes.

  1. Please see the revised abstract.

Case reports.

60, 65, 67. Thank you for pointing out the missing information. We have made the appropriate changes.

  1. The capnograph had the typical waveform of an animal breathing through a Bain system or a circle breathing system, without any evidence of rebreathing or leaks. We have added this information to the manuscript.

  1. We use the term manual ventilation to describe the manual breaths that we delivered. The term controlled mechanical ventilation we use it only once when speaking about the different types of mechanical ventilation, instead of the previously used term IPPV.

  1. Please see the relevant changes in the description of the cases. We had described in the first version of our manuscript that Case No 2 made spontaneous attempts because the NMB was wearing off. We hope that is clearer now.

Case No 2. The dog was breathing spontaneously through a Bain system which changed to a circle just before the atracurium administration, in contrast to case No 1 which was on a non-rebreathing for a short duration of time during mechanical ventilation. We made the necessary changes in the description and the discussion in order to make it clearer.

  1. Only case No 2 showed spontaneous ventilation drive during controlled mechanical ventilation which stopped during manual ventilation when ETCO2 was reduced. Case No 1 had low TOF readings during the anaesthetic and didn’t show any spontaneous drive. A top-up was given 50 minutes after the initial dose.

Discussion

Bullet point 2: We changed the term anatomical dead space to airway dead space. It is suggested in the literature that this term is more appropriate (Mosing et al, 2010). We had decided to use anatomical dead space instead airway dead space as it is most used.

Bullet point 3: Indeed, there must have been leaks in the system-especially in case No 1, that went undetected. As now better described in tables 1 and 2 we can see that there is a discrepancy between inspired and expired Vt (22/14). The difference of 50 and 14 ml doesn’t indicate a leak, as 50 ml was the measured tidal volume on the ventilator. We set the ventilator to deliver 50 ml and was displaying 50 ml on the screen. The spirometer on the other hand was measuring an inspired tidal volume of 22 ml and an expired tidal volume of 14 ml (difference of 8 ml). Part of the discrepancy between the ventilator reading and the “real” inspired tidal volume measured by the spirometer was due to the wasted volume in the tubing (the loss was approximately 28 ml). The difference between inspired and expired tidal volume measured by the spirometer was 8 ml, of which approximately 3 ml were removed during the sampling of gas from the side stream capnograph (aspiration flow rate is 200 ml/min in this type of monitor), and what was left (5 ml) was probably due to a leak, possibly around the ETT. Please see the revised manuscript.

Bullet point 4 and 5: Please see relevant changes in the discussion. The use of the Merlin ventilator on a PCV would have resulted in setting the pressure quite high on the ventilator, in order to deliver enough Vt to ventilate the alveoli. However, the spirometer would read lower (real).

Bullet point 6: Hopefully adding tables 1 and 2 will make clearer the cases’ descriptions.

Mosing, M.; Staub, L. and Moens, Y. Comparison of two different methods for physiologic dead space measurements in ventilated dogs in a clinical setting. Vet Anaesth Analg 2010, 37 (5), 393-400.

Round 2

Reviewer 1 Report

Good resubmission.

Author Response

Thank you very much for your comments. A native English speaker has already proofread the manuscript; however, we have tried to improve it further.

Reviewer 2 Report

Dear authors, thank you for implementing suggestions and improving your manuscript.

My last comment is on your last sentence in the conclusion that I feel doesn't support the message of caution and good practices that need to be tailored to the patient.

Here is your sentence:

"The paediatric lung is more delicate and small errors may have a greater impact. This case report highlights the complexity of ventilating paediatric patients especially without enough training and modern equipment. Emphasis is given to having appropriate monitoring of the ventilated paediatric patient, with the use of spirometry and capnography,
especially when the ventilator used does not have the technology to compensate for the CT. More studies need to be performed on this area of veterinary paediatric mechanical ventilation.

My conclusion while reading your cases and discussion and trying to be in the situations you described is that despite using a ventilator that is designed for a large range of tidal volumes (and therefore various patient's sizes) and extensive respiratory monitoring , espcecially in the case of lack of access to clinical signs of adequate thoacic/abdomen inspiratory excursion and very small patient, the parameters that were used to evaluate the IPPV adequacy and their interpretation failed to detect the complication and it is that precise situation that should have warant early blood gas measurement. We can say, in  an ophtalmologic pratice where every day correlation of clinical signs and monitored parameters is not possible due to the setting with the drapes and the microscope and the interference with the surgery when the anesthetist touch the patient, using advanced respiratory monitoring (spirometry) is considered with good practice and it is reassuring that it is used routinely. However anesthetic preassessment should serve at identifying cases that are not standard and that would benefit from an adaptation of the standard of care and management.

I would expect your conclusion to end on how to avoid the repetion of that scenario, management 'm sure you have already implemented in your pratice.

Thank you again for this great work and being willing to share the experience gained on difficult/unusual cases

Best regards

Author Response

Thank you once again for the constructive comments. We hope that the conclusions are more appropriate after the revision and will add to the prevention of similar complications in the future.

Reviewer 3 Report

Thank you very much for the reviewed manuscript. The information and discussion are now more concise and relevant to the cases. The added tables greatly improve the ease of reading and understanding the cases. If anything, I would consider removing some of the text whenever the tables can be considered enough to explain/report, making the text more succinct and easier/faster to read while still getting the take home message.

Author Response

Thank you very much for your comments.

We understand that we describe details in the text that can be found in the tables too. We decided to do so in order to avoid confusion, especially about the different tidal volumes measured from the ventilator or the spirometer.